# SPARE: A Spectral Peak Recovery Algorithm for PPG Signals Pulsewave Reconstruction in Multimodal Wearable Devices

**DOI:** 10.3390/s21082725

**Published:** 2021-04-13

**Authors:** Giulio Masinelli, Fabio Dell’Agnola, Adriana Arza Valdés, David Atienza

**Affiliations:** Embedded Systems Laboratory, Swiss Federal Institute of Technology in Lausanne (EPFL), 1015 Lausanne, Switzerland; fabio.dellagnola@epfl.ch (F.D.); adriana.arza@epfl.ch (A.A.V.); david.atienza@epfl.ch (D.A.)

**Keywords:** motion artifacts removal, multimodal monitoring, PPG, SPARE, wearables

## Abstract

The photoplethysmographic (PPG) signal is an unobtrusive blood pulsewave measure that has recently gained popularity in the context of the Internet of Things. Even though it is commonly used for heart rate detection, it has been lately employed on multimodal health and wellness monitoring applications. Unfortunately, this signal is prone to motion artifacts, making it almost useless in all situations where a person is not entirely at rest. To overcome this issue, we propose SPARE, a spectral peak recovery algorithm for PPG signals pulsewave reconstruction. Our solution exploits the local semiperiodicity of the pulsewave signal, together with the information about the cardiac rhythm provided by an available simultaneous ECG, to reconstruct its full waveform, even when affected by strong artifacts. The developed algorithm builds on state-of-the-art signal decomposition methods, and integrates novel techniques for signal reconstruction. Experimental results are reported both in the case of PPG signals acquired during physical activity and at rest, but corrupted in a systematic way by synthetic noise. The full PPG waveform reconstruction enables the identification of several health-related features from the signal, showing an improvement of up to 65% in the detection of different biomarkers from PPG signals affected by noise.

## 1. Introduction

The photoplethysmographic (PPG) signal is a noninvasive measure of the blood pulsewaves [1] that reflects the cardiovascular system’s state. For this reason, it allows researchers and clinicians to evaluate various cardiovascular-related diseases, such as atherosclerosis and arterial stiffness [2], and can be even used for biometric identification [3,4]. Moreover, several biomarkers can be extracted from each pulsewave and correlated with cognitive workload, stress, and emotional state of subjects [5,6,7,8,9]. Indeed, these biomarkers allow assessing the physiological changes induced by both physical activity and cognitive tasks (e.g., cardiac response, blood volume, and peripheral blood vessel resistance).

To be specific, the PPG signal is an optically obtained plethysmograph—a measure of the volumetric variations of blood circulation [10]. The principle of operation is the following: thanks to a light-emitting diode (LED) that illuminates the skin, the signal is generated from a photodiode that measures the intensity changes in the reflected light due to the absorption of oxygenated red globules in the blood. As mentioned, several health-related information can be determined. For example, the PPG signal periodicity corresponds to the cardiac rhythm—the so-called pulse interval—and the analysis of each pulsewave can extract features correlated to blood volume, wall vessel elasticity, blood flow velocity, and ankle–brachial index [11,12].

Given its unobtrusive and inexpensive optical measurement, the PPG signal has gained much popularity and has been introduced in many wearable devices [13,14,15,16]. Unfortunately, it is widely used just for heart rate detection, losing most of the valuable information that can be obtained. Indeed, the main drawback that prevents the extraction of nothing more than the cardiac rhythm from PPG-based monitoring techniques is their strong imprecisions during physical exercises and even light daily activities.

In fact, PPG signals—especially the ones obtained from wrist-type sensors—are susceptible to motion. In particular, during exercise, extremely strong motion artifacts (MA) caused by hand movements can contaminate the signal. Figure 1 shows an example of PPG signals of a subject at rest (a), during physical exercise (c), and the corresponding single-sided spectra (b,d). In the spectrum (d), the peak due to the heart rate is highlighted and shows a lower amplitude compared to the peaks corresponding to MA.

This problem is quite well known, and several strategies have been proposed to overcome it [17,18,19,20,21,22,23,24,25,26]. Those techniques share in common the idea of identifying which component of the noisy PPG signals relates to the cardiac rhythm—either using the information from the accelerometer data or employing advanced tracking methods—and use this information for heart rate detection.

Given the many health- and wellness-related information the PPG signal can provide, we believe that its full potential has not been fully exploited yet. There is a lack of MA removal techniques that preserve the full PPG waveform to allow multiparametric monitoring of subjects during daily life and physical activity.

Toward this aim, in this paper, we present a novel method that can reconstruct the entire blood pulsewaves regardless of the motion of the subject by taking advantage of the local physical properties of the signal and the simultaneous availability of the electrocardiogram (ECG) data given by multimodal wearable sensors. Extending and enhancing the concepts within existing works from the evaluation of the signal spectrum [23], the presented technique goes further, aiming at locating and cleaning up from noise the spectral contents related to the whole pulsewaves—not just to the cardiac rhythm. The designed model for full pulsewave reconstruction—by leveraging quasiperiodic PPG signals’ fundamental properties—allows a new interpretation of the photoplethysmographic signals, introducing new paradigms for PPG data compression (and successive reconstruction with almost no information loss) and analysis.

The rest of the paper is organized as follows. Section 2 gives an overview of the state-of-the-art techniques for motion artifact removal from PPG. Section 3 explains the theoretical background of the developed algorithm and the signal’s physical characteristics exploited for its reconstruction. Section 4 describes the developed algorithm, including details on signal processing. Section 5 presents the used methodology for performance assessment. The experimental results for the case of signals acquired during intense physical activity and at rest artificially corrupted by synthetic noise are reported in Section 6. Finally, in Section 8, we draw the main conclusions of this work.

## 2. State-of-the-Art Techniques for Motion Artifact Removal from PPG

In the literature, several techniques have been proposed for MA removal from PPG signals. Specifically, two main families of algorithms can be found: the ones that use the acceleration signal and the ones that do not [17,18,27].

The methods that avoid the use of the acceleration data analyze the statistical information present in the data to distinguish between clean and motion-corrupted data [17,18,28]. Some of those rely on the fact that the high-order standardized statistical moments—e.g., skewness and kurtosis—remain constant in case there is no noise affecting the signal [17]. Thus, by keeping track of the changes in the statistical moments, they can classify parts of the signal as corrupt. The clear disadvantage is that those techniques can only discard the corrupted portions of the signal, without the possibility of recovering any information.

Independent component analysis (ICA) can overcome this issue [19,20], allowing the splitting of PPG signals into several additive subcomponents. However, ICA necessitates several PPG sensors, preventing its application in many wearable devices. Moreover, the subcomponents to be extracted from the signal have to be statistically independent, which might not be the case for MA-corrupted PPG signals, thus limiting its performance in real scenarios.

Finally, several techniques [29,30] aim to generate a reference signal from the corrupted PPG signal using empirical mode decomposition (EMD) or adaptive noise cancellation (ANC) [31]. However, this reference signal can be complicated to reconstruct when the subject is exercising or moving, in general.

It is worth mentioning that most of the techniques that do not use accelerometer data have been designed for clinical purposes where the subject movement is much reduced—most of the time, limited to finger movements [20,21,32] or running [22] (so they are limited to periodic movements at most). In these scenarios, MA cannot be strong, or it can be easily detected (e.g., in the case of periodic physical exercises). Thus, these techniques may not be suitable during intensive physical movements and daily life conditions.

In contrast, the techniques that adopt the accelerometer data have explicitly been designed for this purpose, allowing intense physical activity tracking [23,24,26]. These solutions are mainly based on the PPG signal’s spectral analysis to try to identify—and successively remove—the frequency components associated with the motion artifacts estimated from movement data [24,25,33]. The acceleration data can also be used—with a Kalman filter—to estimate the additive error signal affecting the PPG [25].

One of the most popular techniques that employ motion-related data is the TROIKA framework [23]. Specifically, TROIKA aims at splitting the PPG signal into additive components, whose spectrum is sparse and whose sum is the original signal. The components are then individually analyzed and their spectra are compared with the data from the accelerometers. Finally, the ones considered not to be associated with the heart rate (HR) are then removed to be able to reconstruct a PPG signal where the cardiac rhythm is easily identifiable. Unfortunately, the removal of entire components can be a waste of useful data. Indeed, even though some of the removed components present dominant frequency components due to the subject’s movement, they can also include part of the signal to reconstruct. Moreover, there could be more noise due to motion and sensor displacement than the one correlated with the accelerometric data—for mainly two reasons. First, a spectral analysis—such as the one performed in TROIKA—can only help in the case of periodic movements. Second, the acceleration data (in three axes) reflects the sensor’s movement in space, whereas MA in a PPG signal originated from changes in the gap between skin and the sensor [23]. Consequently, in the case of irregular and sudden hand movements, the mere analysis of the acceleration data may not be enough to remove MA or might not provide substantial benefits.

For this reason, we propose a novel method: spectral peak recovery (SPARE) capable of removing not only periodic movements but also sudden ones. To this aim, we use some of the key processing techniques from TROIKA [23], such as decomposition and sparsity-based high-resolution spectral estimation, and propose a new methodology with the goal of reconstructing the full pulsewave without just focusing on the HR. To fulfill this task, SPARE takes advantage of the availability of simultaneous ECG data provided by new wearable devices to detect the PPG frequency components related to HR reliably. Finally, the full reconstruction of the entire PPG signal allows the extraction of several health- and wellness-related biomarkers.

## 3. SPARE: Theoretical Background

Our proposed SPARE algorithm aims to recover the PPG signal’s main sinusoidal components even if buried in noise. To be specific, the SPARE algorithm exploits several of the crucial elements presented on the state-of-the-art methods and a fundamental property of periodic signals. In particular, our algorithm is inspired by the TROIKA framework, but it looks at the problem from a different perspective. TROIKA assumes to be able to detect the noise (from accelerometers) and tries to remove it; SPARE, on the contrary, estimates the main components of the PPG using the mean HR detected from ECG as reference and extracts them from the noisy signal.

In fact, the main point of the entire algorithm is based on the idea that a periodic signal y(t) with period $T_{0}$ can be written as the sum of sinusoidal components whose frequencies are integer multiples of $\frac{1}{T_{0}}$ (the so-called fundamental frequency):(1)$$y\left(t\right)=a_{0}+\sum_{n=1}^{\infty }a_{n}\cos\left(2\pi \frac{n}{T_{0}}t\right)+\sum_{n=1}^{\infty }b_{n}\sin\left(2\pi \frac{n}{T_{0}}t\right),$$
where $a_{n}$ and $b_{n}$ (for $n\in N^{+}$) constitute the weight of the sinusoidal component whose frequency is $\frac{n}{T_{0}}$. In other words, it is possible to reconstruct the signal y(t) by summing together its sinusoidal components.

Theoretically, the full reconstruction is only possible by adding up all the infinitely many components. However, given the fact that most of the biosignals’ spectral contents are limited to the low frequency range, a proper reconstruction can be obtained by adding back just a few of the first components—only three of them, to be exact. In fact, in the available data sets, we observe that the PPG signal can be gratifyingly reconstructed by only considering the first three components: the fundamental, the second harmonic, and the third harmonic. Giving a rest HR of 80 BPM, the PPG’s fourth harmonic is located at 320 BPM (5 Hz) and—if present—would be already eliminated by the preliminary bandpass filtering (see Section 4 for more details).

An example of the reconstruction of a PPG signal (and the spectrum of the reconstructed signal) keeping nothing but its fundamental frequency, second harmonic, and the third one, is presented in Figure 2.

The choice of the duration of the time segmentation window in which the signal is processed is critical. We need to choose a considerable time span to increase the resolution in frequency. However, the changes in the cardiac rhythm in a long window make the PPG signal aperiodic. Thus, a long window would introduce several frequencies in the spectrum within the interval of variation of the heart rate. In our algorithm, the window length has been set to 8 s, a value that is accepted to be optimal in literature [23,26,34,35,36]. This choice allows an excellent spectral estimation without affecting much the periodicity of the signal.

## 4. SPARE: Spectral Peak Recovery Algorithm

SPARE can be divided into four stages: signal decomposition, spectral estimation, harmonic relationship estimation, and reconstruction. From the first two stages, the spectral estimation of an 8-s-signal segment is obtained to then find and extract its three fist harmonics corresponding to the cardiac activity. Finally, the signal segment is fully reconstructed. Figure 3 represents the complete diagram of our algorithm.

First of all, the raw PPG signals are bandpass-filtered from 0.5 Hz to 5 Hz before any further processing to exclude frequency components that are not physically possible correlated with the PPG data [37,38,39,40,41]. For this step, we used minimum-order filters with a stopband attenuation of 60 dB and compensation for the delay introduced by the filter. Then, we evaluate the quality of the signal to apply SPARE only if it needs to be cleaned. The signal quality assessment employs a trend-based approach that searches for regularity, the fourth standardized statistical moment (kurtosis) [42]. If the kurtosis of the signal in the current window is less than a threshold (fixed at 3, the kurtosis of any univariate normal distribution [43]), no further processing is executed on the signal itself, and it is outputted directly.

After the preliminary filter and signal quality verification, the singular spectrum analysis (SSA) algorithm (see Section 4.1) is used to decompose the signal into its additive oscillatory components (signal decomposition). Second, the sparse signal reconstruction (SSR) technique (see Section 4.2) is used to obtain two spectra: the spectrum of the PPG signal for spectral peak position detection—in the current window—and the one of the second-order temporal differentiation of the signal itself for spectral peak width estimation (spectral estimation). Due to SSR’s high complexity, we also propose an alternative version of the algorithm that uses—for the two spectra—a more common spectral evaluation technique based on fast Fourier transform (FFT). In the remainder of the article, we refer to this variant as fastSPARE.

Next, we extract the location of the dominant peaks in the spectrum to determine which of them corresponds to the fundamental heartbeat frequency of the PPG signal, to the second harmonic, and to the third one—using as reference the HR estimation from the ECG signal. Then, we estimate the width of these three main peaks using the spectrum unaffected by the temporal differentiation (harmonic relation estimation). Finally, with three narrow bandpass filters, we extract nothing but the signal’s three harmonic components (reconstruction).

### 4.1. Signal Decomposition

This first step is derived from [23], where the singular spectrum analysis (SSA) is used to decompose the photoplethysmographic (PPG) signal into *g* oscillatory components.

#### Singular Spectrum Analysis

The adopted implementation of SSA strictly follows the description in [44], so only the main steps are reported. Given a time series $y={\left[y_{1},y_{2},\ldots ,y_{M}\right]}^{T}$, SSA aims at decomposing it in *g* oscillatory components and noise, such as:(2)$$y=\sum_{i=1}^{g}z_{i}.$$

The decomposition’s primary purpose is to allow the identification of the components associated with noise and remove them from (Equation 2). It consists of two complementary stages: decomposition (i) (made up of embeddings (a), singular value decomposition (b), and grouping (c)), and reconstruction (ii). Figure 4 shows an example of the decomposition of a PPG signal.

(i)Decomposition(a)Embeddings Step: The L-trajectory matrix X is created from y. X$\in R^{L\times K}$ (where K=M−L+1, *M* is the length of the time series y, and $L<\frac{M}{2}$ is defined by the user, preferably close to $\frac{M}{2}$), and it is defined as:
(3)$$X=\left|\begin{array}{cccc} y_{1} & y_{2} & \cdots & y_{K}\\ y_{2} & y_{3} & \cdots & y_{K+1}\\ \vdots & \vdots & \ddots & \vdots \\ y_{L} & y_{L+1} & \cdots & y_{M} \end{array}\right|.$$(b)Singular Value Decomposition Step (SVD): The SVD of the matrix $X\in R^{L\times K}$ is given by:
(4)$$X=U\Sigma V^{T}=\sum_{i=1}^{d}\sigma_{i}u_{i}v_{i}^{T}=\sum_{i=1}^{d}X_{i},$$
where d=rank(X)≤min{L,K}, $\sigma_{i}$, $u_{i}$, and $v_{i}$ are the $i^{th}$ singular value, the corresponding left-singular vector, and the corresponding right-singular vector of X, respectively. The collection ($\sigma_{i}$, $u_{i}$, $v_{i}$) is called the eigentriple. Given the fact that the eigentriples whose singular value is close to zero contain no significant information, they can be safely excluded from the reconstruction. In this case, limiting the analysis to the eigentriples corresponding to the biggest 10 singular values allows reducing the computational time without significantly affecting signal reconstruction.(c)Grouping Step: In this step, the *d* rank-one matrices $X_{i}$ are clustered in *g* (<d) groups according to some clustering criterion (such as strong harmonic relation; in this case, if the corresponding singular values are close enough) such that:
(5)$$X=\sum_{j=1}^{g}X_{I_{j}},$$
where $X_{I_{j}}=\sum_{t\in I_{j}}X_{t}$ and $I_{j}$ is the subset of the index set {1,…,d} relative to indices of the matrices $X_{i}$ belonging to the group *j* (ranging from 1 to *g*).(ii)Reconstruction StepIn this phase, each matrix $X_{I_{j}}$ is used to reconstruct a new time series $z_{j}$ of length M using the so-called diagonal averaging procedure [44]. Indeed, let $diag_{avg}\left(A\right)$ be the operator applying the diagonal averaging to a generic matrix A; we have $y=diag_{avg}\left(X\right)$ by construction. Moreover, because of the associative propriety of the addition, we finally obtain *g* new time series $z_{j}$:
(6)$$y=diag_{avg}\left(X\right)=diag_{avg}\left(\sum_{j=1}^{g}X_{I_{j}}\right)=\sum_{j=1}^{g}diag_{avg}\left(X_{I_{j}}\right)=\sum_{j=1}^{g}z_{j}.$$

### 4.2. Spectral Estimation

To be able to obtain a high-resolution spectral estimation of the PPG signal, sparse signal reconstruction (SSR) is applied to each of the *g* additive oscillatory components into which the signal has been decomposed. SSR has been chosen due to its robustness to noise interference compared to traditional nonparametric spectral estimation algorithms. As reported in [45], SSR has a number of advantages over conventional spectral estimators, such as higher spectral resolution and low variance. One of the drawbacks of the SSR that limits its usage in general applications is the requirement of sparsity for the spectrum to estimate. However, in the considered scenario, this aspect does not constitute a limitation since the decomposition performed by SSA has the additional benefit of making the spectrum of the single component sparse.

SSR is part of a family of algorithms that goes under the name of compressive sensing: an approach that exploits the sparsity property as a precondition for signal recovery. Sparse signals are characterized by a few nonzero coefficients in one of their transformation domains. Such property allows their complete reconstructed from a reduced set of available measurements.

SSR aims at recovering a sparse vector x that solves the following equation:(7)y=Φx+ν.
where y is an observed signal of length M, Φ a known matrix of size M×N representing the linear transformation undergone by the sparse signal that we are going to recover, and ν an unknown noise vector.

An estimator of the solution of the previous equation (Equation 7) can be obtained by solving the following optimization problem:(8)$$\hat{x}=\underset{x}{arg min}{∥y-\Phi x∥}_{2}^{2}+\lambda g\left(x\right),$$
where λ∈R is a regularization parameter and g(x) is a penalty function enforcing sparsity on the solution (for example, ${∥x∥}_{1}$). Conveniently, by choosing the (m,n)th element of Φ to be $e^{j\frac{2\pi }{N}mn}$, for m=0,⋯,M−1 and n=0,⋯,N−1, the solution of (Equation 8) leads to the sparse spectrum of y.

The application of SSR to each of the *g* additive components leads to *g* spectral estimates. In this case, instead of computing the spectral peaks for every single oscillatory component, as it is done in [23], we obtain the original signal spectrum by summing up the individual spectra leveraging on the linearity of the transformation. Let $s_{orig}$ be the just computed spectrum.

The same procedure is then repeated with the addition of the second-order temporal differentiation of the *g* additive components before feeding them to the SSR algorithm. The temporal difference is the operation that returns the difference between consecutive values of a time series. In practice, given $y={\left[y_{1},y_{2},\cdots ,y_{M}\right]}^{T}$, its first order temporal differentiation gives: $y^{\prime }={\left[y_{2}-y_{1},y_{3}-y_{2},\cdots ,y_{M-1}-y_{M}\right]}^{T}$. In general, the *k*th order difference preserves the original signal’s harmonic content—although altering the “timbre” of the signal by linearly increasing the amplitude of the harmonics with frequency—while ignoring the aperiodic portions of the signal. The advantage is that we can eliminate all the aperiodic components that may be contained, and that can lead to the wrong recognition of the spectral peak associated with the cardiac rhythm. Unfortunately, it can also slightly alter the informative content of the spectrum. For this reason, whenever we need to extract precise information related to the frequency of the peaks or their width, we consider the original spectrum $s_{orig}$. In the remainder of the article, we refer to the temporally differentiated spectrum as $s_{diff}$.

#### fastSPARE

Since the steps using SSR require the solution of an optimization problem (see Section 4.2), the algorithm’s computational time is quite long. This inconvenience motivates us to develop an alternative version of SPARE, where all the spectral estimation problems have been replaced by regular FFTs (see Figure 3 for the comparison of the FFT-based version with the original one). The massive speed-up (computational time for a 20 min long signal has been reduced from almost 2 h to 30 s, on a regular PC) could potentially allow the real-time implementation of the algorithm. We refer to this faster variant as fastSPARE. In the following section, we are going to analyze its performance compared to the one of the original SPARE.

### 4.3. Harmonic Relation Estimation

In this stage, $s_{diff}$ (see Section 4.2) is used to extract the three harmonics of the PPG signal (in the considered window), using as reference the mean heart rate (meanHR) obtained from the ECG data. Then, from $s_{orig}$—the one not affected by temporal differentiation (see Section 4.2)—we estimate the width of these three main harmonic peaks, which define the frequency components that are used to reconstruct the signal.

From $s_{diff}$, first, we extract the location in frequency of all the peaks in the spectrum. Then, from this set of peaks, we identify three subsets, namely $Fund_{cand}$, $Second_{cand}$, and $Third_{cand}$. These are the set of peaks whose frequency is less than 10 BPM away from meanHR, the set of *p* peaks whose frequency is closer to two times meanHR, and the set of *p* peaks whose frequency is closer to three times meanHR, respectively. For robustness, from $Fund_{cand}$, we remove the peaks whose amplitude is less than a percentage (20%) of the highest peak in the same set. Figure 5 shows an example of these three sets where *p* is set to 3. Next, we chose $HR_{cand}\in Fund_{cand}$, $HR_{cand_{2}}\in Second_{cand}$, and $HR_{cand_{3}}\in Third_{cand}$ such that they are in harmonic relationship, i.e., we look for the triplet from the three sets that better approximate the following relations:(9)$$\frac{HR_{cand_{2}}}{HR_{cand}}=2$$
(10)$$\frac{HR_{cand_{3}}}{HR_{cand}}=3.$$
In case no triplet satisfies (Equation 9) and (Equation 10) within a tolerance ($|\frac{HR_{cand_{2}}}{HR_{cand}}-2|\leq 0.3$), $HR_{cand}$ is set to meanHR, and $HR_{cand_{2}}$ and $HR_{cand_{3}}$ are computed to satisfy (Equation 9) and (Equation 10). Once the frequency locations of the fundamental, the second harmonic, and the third one have been located, the width of the three peaks is estimated by looking at the zero crossing of the derivate of the spectrum $s_{orig}$ on the right and on the left of the peak location.

### 4.4. Reconstruction

For each of the identified three components, we use a narrow bandpass filter, with the width previously computed. For this step, we used minimum-order filters with a stopband attenuation of 60 dB and compensation for the delay introduced by the filter. Finally, adding the so-obtained three signals, we recover the PPG signal (see Figure 6 for an example).

## 5. SPARE Performance and Robustness Assessment

SPARE is meant to reconstruct the PPG signal’s full waveform even when strong motion artifacts are present. Specifically, its ultimate goal is to make the following delineation algorithm work by making the full PPG waveform available by the algorithms that extract its characteristic point, regardless of the subject movements. Figure 7 shows the principal fiducial points from which PPG biomarkers are extracted.

To assess SPARE performance and robustness, an annotated reference PPG signal is needed; however, there are no such available data. Thus, we evaluated the algorithm performance using a clean PPG signal that we artificially corrupted by synthetic noise.

In doing so, we have both a reference PPG signal that can be used as ground truth for evaluating the reconstruction error, and reference fiducial points obtained by running the delineation algorithm on the clean signal to test the delineation accuracy. For this purpose, we used a data set that includes PPG signals acquired at rest during a relax session (i.e., almost not corrupted by MA) [46], where we incorporated different levels of synthetic noise to assess the robustness of SPARE to different types of MA.

To also test the algorithm on PPG data affected by “natural” MA, we also use two additional data sets containing synchronous PPG and ECG data affected by subject movements. In these cases, we can only test the HR detection since there is no other ground truth reference, even though SPARE is able to extract additional cardiovascular information (see Figure 7.

The second data set includes data collected during an experimental session where subjects simulate manual labor (with asynchronous and sudden movements) [9]. Similarly, the third database (from [23]) includes data from subjects walking or running on a treadmill. This database has been widely used to assess different MA reduction techniques [36,47,48] by comparing the average HR detection on 8-s-windows obtained from PPG with the reference HR obtained from the time intervals in between one beat to another (RR intervals) computed from ECG.

This strategy—using the first database—allows us not only to determine how the noisy PPG signals processed by SPARE are close to the reference but also the detection accuracy for four fiducial points—using the delineation of the noise-free signals as ground truth. Finally, with the other two data sets, we can also test the HR detection since we believe it to be beneficial to evaluate the overall performance of the algorithm, even though it is not the primary goal of SPARE.

### 5.1. Evaluation on PPG Signals Corrupted by Adding Synthetic Noise

In order to evaluate the SPARE capabilities in terms of quality of the reconstruction of the PPG signals and detection accuracy of multiple fiducial points, we applied the algorithm to an artificially corrupted reference PPG signal acquired from a subject at rest and compared its output to the original signal. The PPG and ECG signals were sampled at 250 Hz using a Medicom device, ABP-10 module (Medicom MTD Ltd., Russia) during a relax session (see [46] for more details.) To be specific, we consider the ability to delineate multiple fiducial points from the signals as a metric for evaluating the signal quality [49]. Thus, the delineation of the SPARE-filtered signal is compared with the delineation of the original signal to assess how SPARE can improve the detection accuracy of several fiducial points in presence of noise. The evaluation includes the following metrics: sensitivity, predictivity (positive predictive value), and geometric mean between the last two. Moreover, we also evaluate the mean squared error between between the original PPG signal and the reconstructed one by SPARE and fastSPARE.

To artificially corrupt the reference signals, we used a novel noise generator. In fact, conventional noise generators—using random noise drawn from different distributions such as Gaussian or Poissonian—do not allow to properly evaluate the algorithm’s performance, as they can only provide unrealistic noises compared to the one commonly found in PPG signals. Indeed, MA generally appears in the form of sudden spikes (in correspondence to the subject’s movement) and slowly varying offsets (baseline wander) due to the changes in distance between the skin and the sensor after every sudden movement. To overcome this issue, we designed a more realistic synthetic noise generator that can simulate those two behaviors, enabling us to corrupt a reference signal with different noise levels.

#### Realistic Synthetic Noise Generator

The motion artifacts that affect the PPG signals present sharp spikes alternated by periods with more gentle variations. To obtain a realistic noise, therefore, we combined conventional random noises with signal interpolation techniques.

To be specific, given a Lanczos kernel L(n) made up of T samples such that:(11)$$L\left(n\right)=\left\{\begin{array}{cc} sinc\left(n\right)sinc\left(\frac{n}{T}\right) & \mathrm{if}\;-\frac{T+1}{2}+1\leq n\leq \frac{T+1}{2}\\ 0 & \mathrm{otherwise} \end{array}\right.,$$
and a random signal $z={\left[z_{1},z_{2},...\right]}^{T},$ where $z_{i}$ for i=1,2,... is sampled from a zero-mean normal distribution with standard deviation (σ), the noise is obtained as: (12)$$N\left(n\right)=\sum_{i=⎡\frac{n}{stride}⎦}^{⎡\frac{n}{stride}⎦+\frac{T}{stride}-1}z_{i}L\left(n-i\times stride+T-1\right),$$
where ⎡x⎦ denotes the ceiling function that maps x∈R to the least integer greater than or equal to *x*. stride represents the number of samples by which each kernel is shifted for the construction of the noise, and $\frac{T}{stride}\in N^{+}$.

In particular, if the used kernel size *T* is relatively big and if the kernel’s last and first samples are not close to zero, this method allows generating a noise that gently varies whenever the kernels do not overlap on their boundaries and becomes spiky when they do. It also gives the possibility to control the noise level by modifying the standard deviation (σ) of the randomly generated signal z.

Figure 8 shows an example of a portion of the reference PPG signal contaminated by our synthetic noise using four different values for sigma, namely, 0 (reference signal), 100, 200, and 500. It also contains an example of a PPG signal naturally affected by noise (the standard deviation of the shown signal is 417). An empirical evaluation on our data sets shows that we can obtain a realistic level of noise during a moderate physical activity using σ values in the range of 400–500.

### 5.2. Evaluation on “Naturally” Corrupted PPG Signal by MA

To evaluate the performance of the algorithm for PPG signals affected by MA, we considered a time window of 8 s with an overlap of 2 s sliding on the extracted RR and PP series (that are the beat-to-beat time intervals in the ECG and PPG signals, respectively). In the said window, we compute the average RR interval from the simultaneous ECG data, the average PP extracted from the signal reconstructed by SPARE, and the average PP extracted from the PPG signal without MA removal (only bandpass filtered between 0.5 Hz and 5 Hz).

The validation relied upon two already existing data sets. The first one encompassed data from twelve subjects that were performing various activities, including tightening some screws in blocks of wood at different heights, walking, and solving arithmetic tasks (about 18 min per subject, ethical approval HREC:037-2019; see [9] for more details about the experimental protocol). The PPG sensor used was an Empatica E4 (with a sampling frequency of 64 Hz) and, for the ECG, a Shimmer3 ECG (with a sampling frequency of 512 Hz). The second database included the data from twelve subjects performing various physical exercises on a treadmill (see [23]). Two-channel PPG signal and three-axis accelerometer were recorded from the subjects’ wrist and one-channel ECG from the subjects’ chest as ground-truth of the heart rate, each sampled at 125 Hz. In our case, we used only one of the two PPG channels.

Finally, to evaluate the performance, we use the mean absolute error ($AE_{mean}$), the mean relative error ($RE_{mean}$), and the median absolute error ($AE_{median}$) per subject, defined as:(13)$$AE_{mean}=\frac{1}{N}\sum_{i=1}^{N}\left|HR_{ECG_{i}}-HR_{SPARE_{i}}\right|,$$
(14)$$RE_{mean}=\frac{1}{N}\sum_{i=1}^{N}\frac{|HR_{ECG_{i}}-HR_{SPARE_{i}}|}{HR_{ECG_{i}}},$$
(15)$$\begin{array}{c} AE_{median}=median\{[|HR_{ECG_{0}}-HR_{SPARE_{0}}|,\\ \ldots ,|HR_{ECG_{N}}-HR_{SPARE_{N}}{\left|\right]}^{T}\}, \end{array}$$
where *N* is the number of the considered windows, $HR_{ECG_{i}}$ is the reference heart rate computed from the ECG data in the *i*th window, and $HR_{SPARE_{i}}$ is the estimated HR from the PPG signal reconstructed by SPARE.

The same procedure is applied to the HR estimated from the PPG without MA removal, $HR_{PPG_{{MA}_{i}}}$, for comparison.

## 6. SPARE Performance Evaluation Results

In this section, we present the results of the performance evaluation of the algorithm in two ways. First, using the developed noise generator, we assessed the detection accuracy of four different fiducial points and the mean square error between the reconstructed signal and the reference one to variation of noise level. Second, we used the data collected during experimental sessions where the subjects’ movements “naturally” corrupted PPG signals from MA. In this case, we analyzed how close the HR estimation from SPARE-filtered signals is to the reference HR computed with ECG data.

### 6.1. Accuracy for Fiducial Points Extraction Using SPARE on Synthetically Corrupted Reference PPG Signals

Figure 9 and Table 1 show the behavior of the geometrical mean between specificity and sensitivity of the delineation algorithms for different noise levels for the following fiducial points: slope, peak, onset, and dicrotic point for both SPARE and fastSPARE. The sensitivity and specificity are computed based on the true positives (TPs), false positives (FPs), and false negatives (FNs) of the fiducial points detected by our delineation algorithm on the signals corrupted by noise against the detection of the same algorithm on the original signal. We used ±0.15 s of tolerance from the reference points. As a comparison, we also included the same results in the case in which SPARE is removed from the processing pipeline, but keeping the initial bandpass filter. SPARE is able to improve the detection accuracy by up to 65%, even in presence of huge quantities of noise (see Figure 8 for an example of noise level up to σ=500).

In general, the detection of all four fiducial points benefits from the introduction of the SPARE filtering, with robust performance that is almost constant regardless of the noise intensity. The baseline technique, conversely, suffers from the introduction of artifacts and reduces its detection accuracy significantly as soon as the standard deviation of the noise increases over 500.

Interestingly, among the four different fiducial points, the dicrotic point appears to be the most challenging marker to detect. As can be seen in Figure 9, even if its accuracy is lower compared to the other markers, its detection is the one that benefits the most from the SPARE reconstruction. Indeed, once the noise’s standard deviation exceeds 200, the detection accuracy without the use of SPARE drops to zero, making this fiducial point almost impossible to detect without SPARE’s aid.

SPARE and fastSPARE appear to perform similarly in this context, with SPARE always in slight advantage over its less-complex counterpart in the order of percentage units. The performance difference depreciates a bit for the dicrotic point detection, which is the hardest fiducial point to be discerned (see Figure 7), even by professionals [50].

Finally, in Figure 10, we report the mean square error (MSE) between the original signal without added noise and the PPG signal reconstructed by SPARE and fastSPARE, as well as without SPARE. As per Figure 10, for both SPARE and fastSPARE, the MSE ranges from 211 (with a noise’s standard deviation of 56) to 1092 (standard deviation of the noise 5000), whereas the MSE for the raw data ranges from 2016 (standard deviation of the noise 56) to 12 M (standard deviation of the noise 5000). This result proves the robustness of our approach; it is able to keep the reconstruction error below 500 until the standard deviation of the noise reaches 2000. In this case, no significant difference—within one standard deviation of each measurement for levels of noise corresponding to a realistic physical activity (standard deviation 100–500)—is appreciable between SPARE and fastSPARE, confirming the value of fastSPARE when it comes to the implementation in applications that requires lower complexities.

### 6.2. HR Estimation Using SPARE on Naturally Corrupted PPG Signals

As previously mentioned, we also evaluated the algorithm’s performance on “naturally” corrupted PPG signals from MA by computing the average pulse period (PP) in a window of 8 s and comparing it to the average RR interval from simultaneous ECG data. As a comparison, we analyzed the impact of SPARE by reporting the results obtained when removing it from the signal processing pipeline.

Table 2 reports the error rates on estimating the HR for the data set from [9] using as reference the HR from the ECG signal after applying SPARE and fastSPARE and the baseline methodology without MA removal. It can be seen that SPARE is able to reduce the mean absolute error ($AE_{mean}$) in HR estimation from PPG by 45.05%. In fact, the average $AE_{mean}$ for the data set here considered is 4.00 BPM for SPARE and 7.28 BPM for the data without MA removal. In this case, fastSPARE outperforms SPARE, with an $AE_{mean}$ of 3.61 BPM (improvement of 50.41% over the baseline technique). The same conclusions also hold for the average relative error ($RE_{mean}$) (that improves by 47.26% using fastSPARE and 40.72% using SPARE), but they do not for the median absolute error ($AE_{median}$) (increment of 46.86% using SPARE and 44.24% using fastSPARE). This behavior can probably be explained by the fact that SPARE’s higher frequency resolution is useful when we have access to high-quality ECG data—for a proper choice of the peaks related to heart rate—but can be detrimental when using noisier ECG (as this is the case). On the contrary, the higher precision allows to decrease the variance of the outcome for different subjects, as confirmed by the lower median error for SPARE compared to fastSPARE.

Even better performance is obtained for the database from [23] that incorporates signals acquired with higher quality sensors affected by mostly periodic movements (see Table 3). SPARE and fastSPARE perform similarly, with $AE_{mean}$ ranging from 0.32 to 1.68 BPM (with an average of 0.78 BPM and 0.72 BPM for SPARE and fastSPARE, respectively). Even though the results are not directly comparable with the state-of-the-art techniques (that have different goals), we achieve an HR detection in the same range (see [51], for which the $AE_{mean}$ ranged from 0.49 to 3.81 BPM averaging 1.29 BPM). This evaluation has been mainly chosen to verify SPARE’s capabilities to handle signals from real subjects, more than to compared it with state-of-the-art HR detection techniques. Indeed, SPARE can make use of the ECG signal, but we have to keep in mind that this signal is used to simplify the detection of the correct triplet that is in turn used to reconstruct the signal. The HR is then detected from the PP series outputted by the delineation algorithm on this reconstructed signal.

This evaluation confirmed that—even though limited to just the HR, not being able to access reference annotated fiducial points and biomarkers in PPG signals—SPARE is useful not only in the presence of repetitive movement (e.g., running) but also when sudden movement occurs while performing manual labor.

## 7. Discussion

Due to its unobtrusive nature and relatively easy implementation in Internet-of-Things devices, the PPG signal dominates wearables’ panorama for heart rate detection. Even so, its usefulness goes beyond the simple detection of the cardiac rhythm. In fact, this signal can be deployed to extract many biomarkers that can be used on multimodal health and wellness monitoring applications. Unfortunately, their simple functioning principle exposes PPG detectors to inaccuracies and disturbs whenever the subject is even slightly moving. Many techniques have been developed to address this issue, but only focusing on the estimation of HR—even employing sophisticated and advanced signal processing pipelines.

In this paper, we aim to exploit the recent advances in state-of-the-art processing algorithms and properties that are fundamentally related to the nature of the PPG signal itself. This approach allows the reconstruction of the signal waveform without additional information from the accelerometers, whose utility might be limited as they only detect the sensor’s absolute movement and not the relative displacement of the detector with respect to the subject skin (that is the source of motion artifacts). Indeed, as confirmed by the experimental evaluation, SPARE was able to reduce the impact of MA on the pulse peak detection accuracy both in the case of intense periodic movements and for sudden ones, while also allowing the detection of several other fiducial points on the pulse waveform.

To the best of our knowledge, there are no available annotated databases that contain noisy PPG signals and a clean reference signal to assess the full waveform reconstruction algorithm. Thus, to evaluate the algorithm’s ability to reconstruct the PPG waveform properly, we relied on our synthetic noise generator that aims at reproducing realistic MA, including both a gently varying offset and rapid spikes. For example, using SPARE with a standard deviation of 450 for the noise (considered to be realistic for moderated physical activity), we obtained a detection accuracy of 93.47%, 89.68%, 89.68%, and 54.84% for slope, peak, onset, and dicrotic point, respectively, corresponding to an improvement of 7.55%, 24.45%, 37.69%, and 49.55% over the same processing pipeline with the exclusion of SPARE.

Whereas the algorithm’s performance looks robust to the varying noise levels, we must recall that its strong MA capability comes at the expense of a high complexity that also requires adopting a multimodal wearable sensor equipped with ECG detection. The use of these two signals—of which one (the ECG) provides a reference for the other (the PPG)—can also create doubt about SPARE helpfulness for real-life applications when both signals might be distorted. However, we must recall that the ECG signal is less prone to motion artifacts compared to the PPG. Indeed, it is relatively simple to extract the only information SPARE needs from it (the HR), given that the so-called R peaks are generally detectable even in the presence of noise. Thus, we believe that the downsides of requiring the ECG signal are fully compensated by the SPARE’s advantage of proving the full reconstruction of the signal, proved to be essential for all the cases where the PPG analysis is not limited to just the HR detection. To reduce the SPARE demand in terms of computation expenses, we also proposed a light-weighted version, more suited to be implemented in real-time on resource-constrained devices.

In summary, the novelty of the suggested approach is that it locally reconstructs the full waveform of the PPG signal just from the three spectral harmonic components related to the cardiac activity. Furthermore, this new interpretation of the PPG signal also allows new paradigms for PPG data compression (and successive reconstruction with almost no information loss) and analysis. For example, it is possible to encode a full 8 s window of PPG data into a condensed representation made up of three complex numbers—representing the phase and the amplitude of the signal’s main three frequency components—and one single scalar (the frequency of the fundamental). Practically, this representation can be stored as seven floating-point values, allowing a memory footprint reduction of 98.6% and 94.53% considering an 8 s window using 16 bits per sample acquired at 125 Hz and 32 Hz, respectively.

## 8. Conclusions

This article presents SPARE, a novel artifact removal technique for PPG that fully reconstructs the PPG entire waveform. We showed that by applying SPARE, we can extract several health- and wellness-related biomarkers, almost regardless of the signals’ noise content. Specifically, SPARE exploits some key physical properties of the PPG signal (for example, its semiperiodicity), the simultaneous availability of ECG data given by multimodal wearable sensors, and state-of-the-art advanced methods for spectral estimation. The introduced methodology also allows new paradigms for PPG data compression (and successive reconstruction with almost no information loss) and analysis. Additionally, we also introduced a noise generator able to simulate and control the motion artifacts’ entity in the signals.

To be able to prove SPARE usefulness, several experimental results are reported both in the case of signals acquired during intense physical activity and at rest that are artificially corrupted by our synthetic noise generator. In our investigations, SPARE provided an improvement of up to 65% for the detection of different biomarkers. Therefore, SPARE provides an excellent reconstruction even in signals that are entirely affected by artifacts allowing the extraction of several valuable biomarkers.

## Figures and Tables

**Figure 1 sensors-21-02725-f001:**
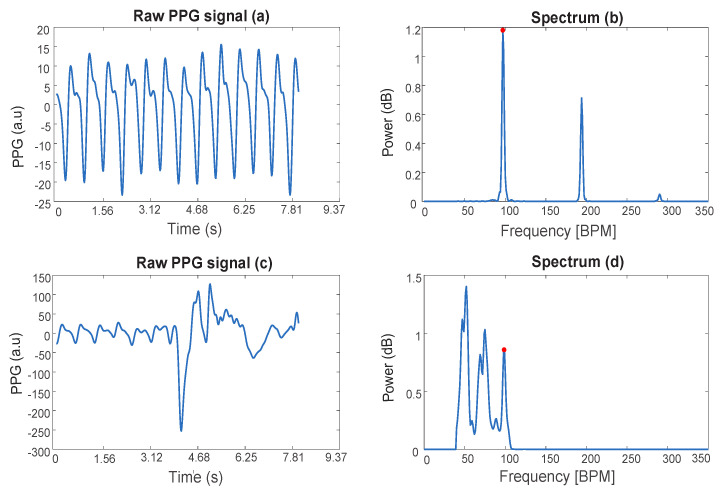
Photoplethysmographic (PPG) signals and corresponding spectra. (**a**,**c**) represent a PPG signal of a subject at rest and performing physical exercises with MA, respectively. (**b**,**d**) show the corresponding single-sided amplitude spectrum. The spectral peak that corresponds to the cardiac rhythm is marked with a red dot.

**Figure 2 sensors-21-02725-f002:**
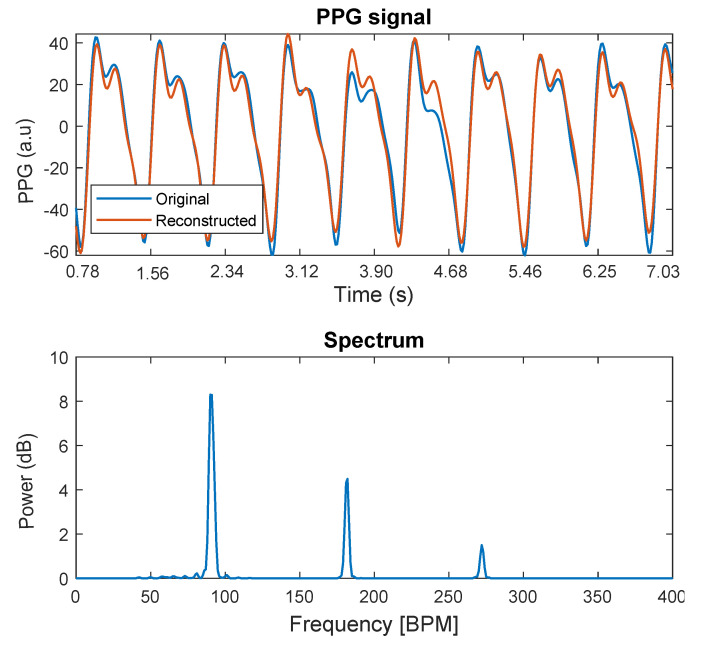
Example of a PPG signal reconstruction (and the spectrum of the reconstructed signal) keeping nothing but its fundamental frequency, second harmonic, and third one.

**Figure 3 sensors-21-02725-f003:**
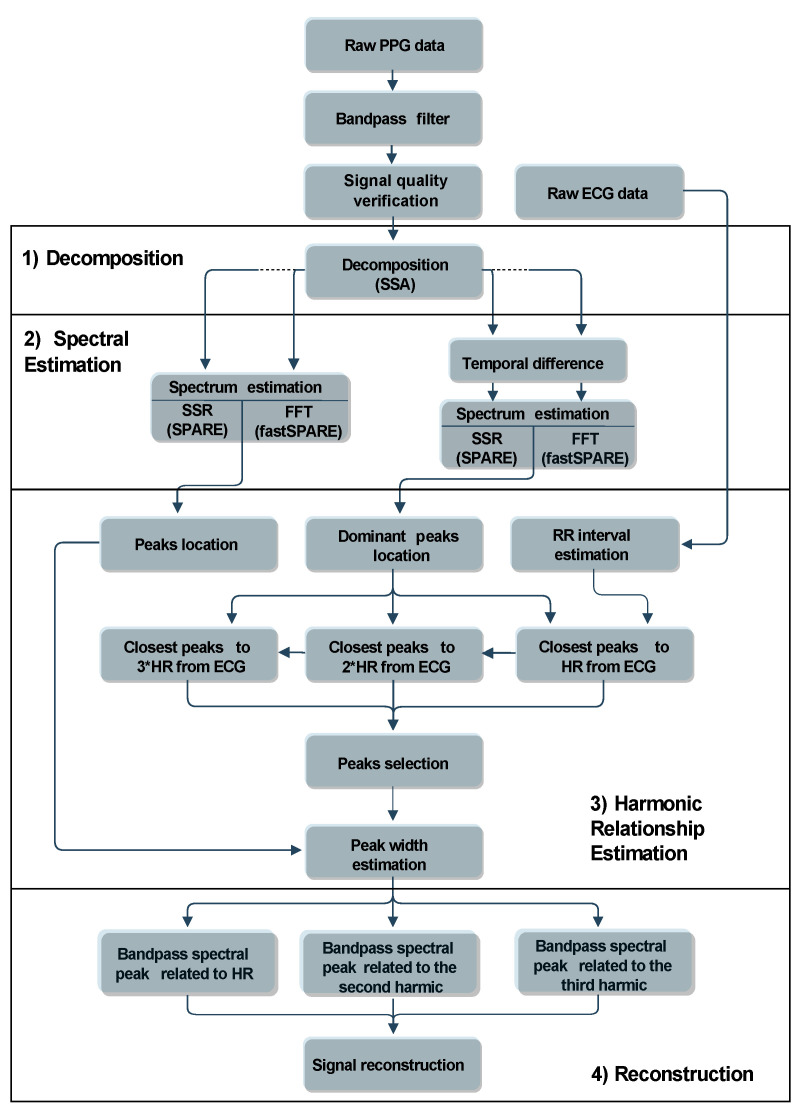
Block diagram of the SPARE algorithm. SSA: singular spectrum analysis, SSR: sparse signal reconstruction, FFT: fast Fourier transform, HR: heart rate, RR interval: beat-to-beat time interval detected from the electrocardiogram (ECG).

**Figure 4 sensors-21-02725-f004:**
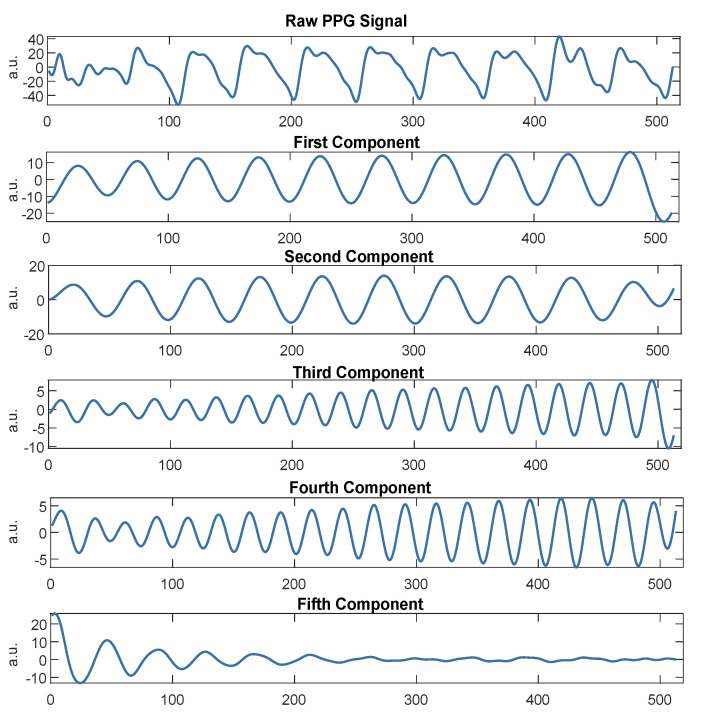
Decomposition of a PPG signal by means of singular spectrum analysis into five additive subcomponents.

**Figure 5 sensors-21-02725-f005:**
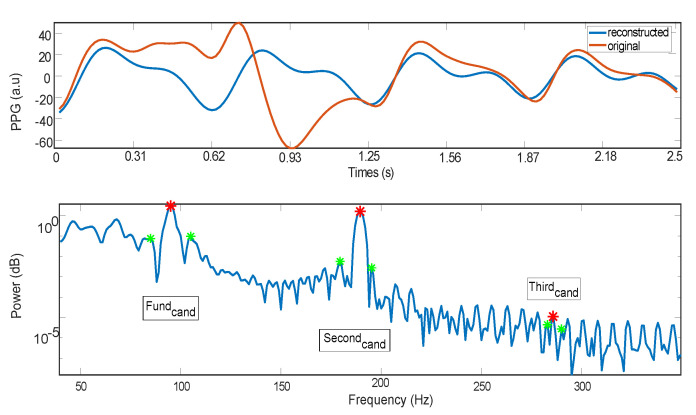
Example of harmonic relation estimation. In green, the candidate peaks for the fundamental, second harmonic, and third harmonic. In red, the detected fundamental, second harmonic, and third one. $Fund_{cand}$: set of peaks whose frequency is less than 10 BPM away from meanHR, $Second_{cand}$: set of *p* peaks whose frequency is closer to two times meanHR, $Third_{cand}$: set of *p* peaks whose frequency is closer to three times meanHR. In this example, *p* has a value of 3.

**Figure 6 sensors-21-02725-f006:**
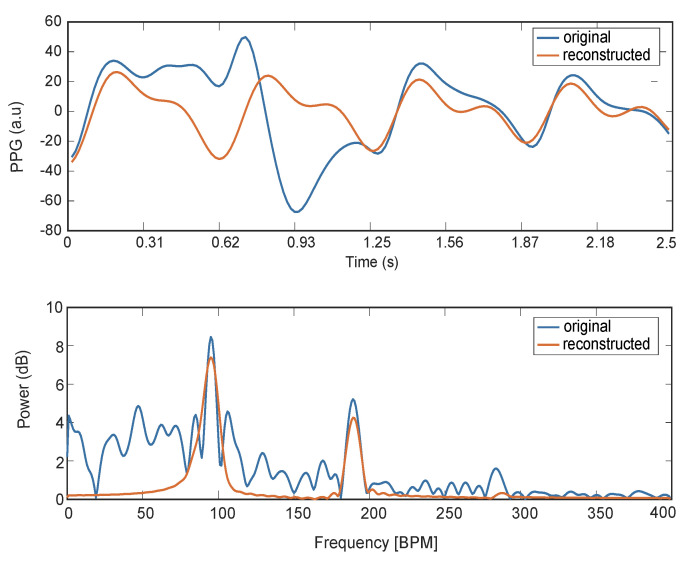
Example of noisy PPG signal and its spectrum. Both original and reconstructed are reported.

**Figure 7 sensors-21-02725-f007:**
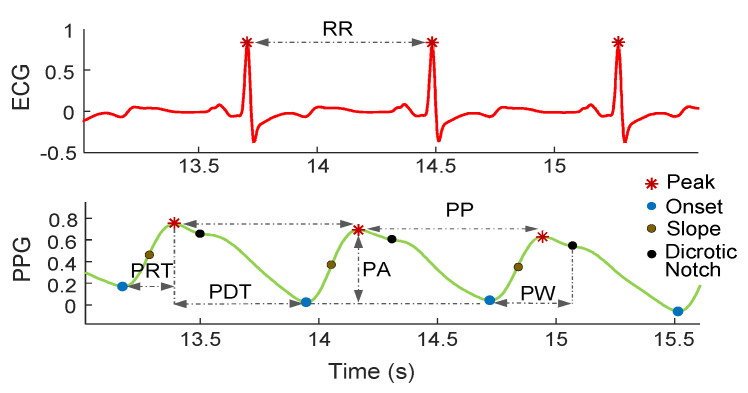
Biomarkers and fiducial points extracted from ECG and PPG signals. PRT: pulse rising time, PDT: pulse decreasing time, PA: pulse amplitude, PP: pulse period, PW: pulse width, RR interval: beat-to-beat time interval detected from the ECG.

**Figure 8 sensors-21-02725-f008:**
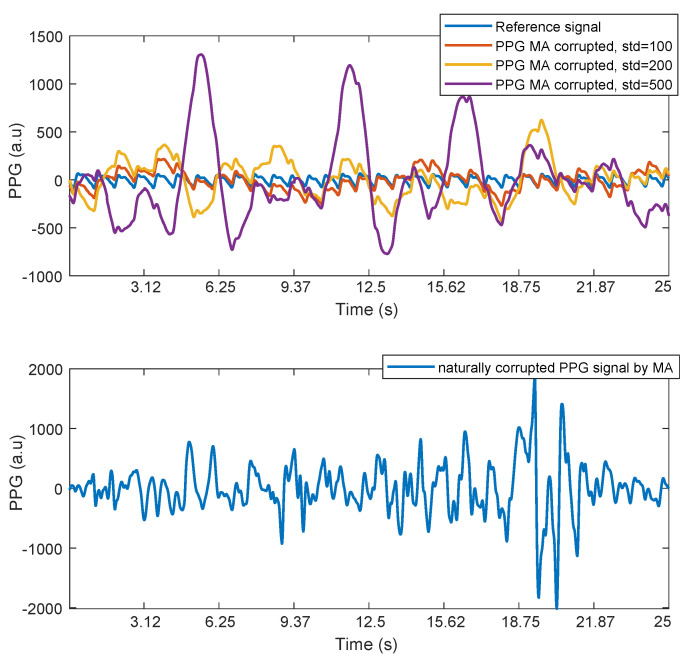
(**Top**) Reference PPG and PPG corrupted by noise using the synthetic noise generator at three different intensity levels (σ = 100, 200, and 500, T=500, and stride=100). (**Bottom**) PPG signal naturally corrupted by noise during physical activity.

**Figure 9 sensors-21-02725-f009:**
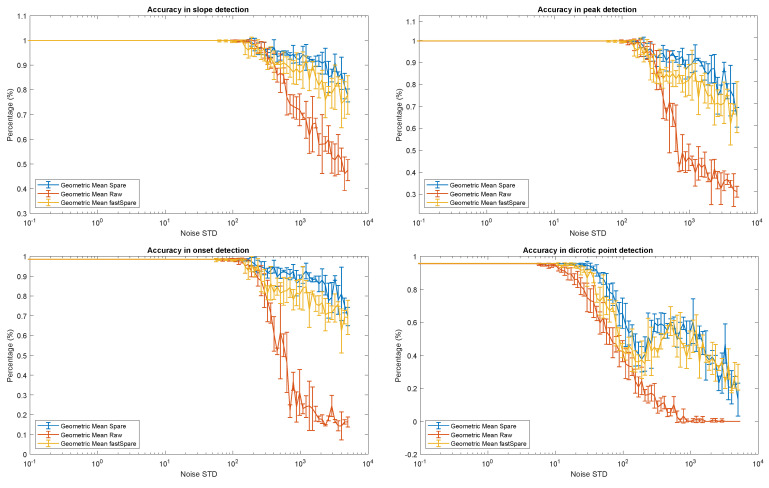
Average and standard deviation of geometric mean between sensitivity and specificity for the detection of slope, peak, onset, and dicrotic points at different noise levels with SPARE (blue line), fastSPARE (yellow line), and raw data (orange line). All experiments were run for five random seeds each (for noise generation).

**Figure 10 sensors-21-02725-f010:**
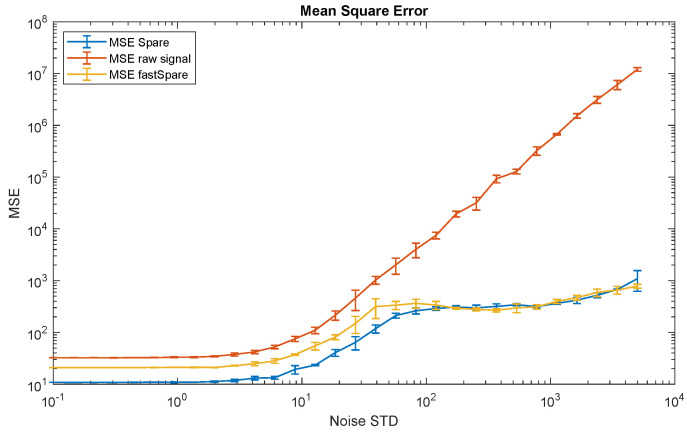
Mean square error (MSE) between the original PPG signal and the reconstructed one by SPARE and fastSPARE at different noise levels. For comparison, the MSE between the PPG signal without SPARE and the reference is also reported (orange line).

**Table 1 sensors-21-02725-t001:** Evaluation of SPARE and fastSPARE in terms of geometric mean between sensitivity and specificity for four different fiducial points: slope, peak, onset, and dicrotic point for different noise levels. In brackets, the increment with respect to the same processing pipeline but excluding SPARE (applying the delineation algorithm to the signal after a passband filter).

NoiseSTD (σ)	Fiducial Points Detection Gmean [Increment over Baseline due to SPARE] (%)															
	Slope				Peak				Onset				Dicrotic Point			
	SPARE		fastSPARE		SPARE		fastSPARE		SPARE		fastSPARE		SPARE		fastSPARE	
0	100	[0.00]	100	[0.00]	100	[0.00]	100	[0.00]	100	[0.00]	100	[0.00]	100	[0.00]	100	[0.00]
0.5	100	[0.00]	100	[0.00]	100	[0.00]	100	[0.00]	100	[0.00]	100	[0.00]	100	[0.00]	100	[0.00]
1	100	[0.00]	100	[0.00]	100	[0.00]	100	[0.00]	100	[0.00]	100	[0.00]	100	[0.00]	100	[0.00]
4	100	[0.00]	100	[0.00]	100	[0.00]	100	[0.00]	100	[0.00]	100	[0.00]	100	[0.00]	100	[0.00]
12	100	[0.00]	100	[0.00]	100	[0.00]	100	[0.00]	100	[0.00]	100	[0.00]	100	[8.21]	93.90	[2.11]
40	100	[0.00]	100	[0.00]	100	[0.00]	100	[0.00]	100	[0.00]	100	[0.00]	91.49	[26.12]	74.65	[9.28]
130	99.93	[0.08]	99.77	[−0.08]	98.43	[0.37]	98.36	[0.30]	98.29	[0.38]	98.36	[0.45]	55.04	[21.91]	43.11	[9.98]
450	93.47	[7.55]	90.65	[4.73]	89.68	[24.45]	83.33	[18.10]	89.68	[37.69]	84.71	[32.72]	54.84	[49.55]	55.86	[50.57]
1500	91.83	[25.71]	89.77	[23.65]	88.10	[47.18]	81.30	[40.38]	88.10	[65.06]	81.73	[58.69]	47.50	[46.42]	48.46	[47.38]
5000	77.70	[30.19]	77.89	[30.38]	63.70	[33.85]	68.38	[38.53]	69.61	[53.26]	69.18	[52.83]	13.17	[13.17]	26.80	[26.80]

**Table 2 sensors-21-02725-t002:** Evaluation of SPARE and fastSPARE by comparing heart rate error (BPM) from ECG (RR-intervals) and PPG (PP-intervals) on windows of 8 s.

Subject	fastSPARE			SPARE			Data without MA Removal		
	${\mathrm{AE}}_{\mathrm{mean}}$	${\mathrm{RE}}_{\mathrm{mean}}$	${\mathrm{AE}}_{\mathrm{median}}$	${\mathrm{AE}}_{\mathrm{mean}}$	${\mathrm{RE}}_{\mathrm{mean}}$	${\mathrm{AE}}_{\mathrm{median}}$	${\mathrm{AE}}_{\mathrm{mean}}$	${\mathrm{RE}}_{\mathrm{mean}}$	${\mathrm{AE}}_{\mathrm{median}}$
S1	3.80	4.37%	1.38	6.26	7.41%	1.50	6.44	7.13%	2.23
S2	8.69	10.46%	5.33	6.91	8.57%	3.17	8.41	9.77%	6.11
S3	3.29	3.13%	2.30	3.22	3.18%	2.39	10.77	9.86%	6.60
S4	3.30	3.52%	2.20	3.19	3.44%	2.08	8.06	8.12%	5.13
S5	4.31	4.94%	1.78	6.69	7.61%	2.21	7.05	7.89%	3.48
S6	3.44	3.90%	2.22	5.22	5.96%	2.15	7.03	7.83%	3.42
S7	2.62	2.77%	1.54	2.58	2.72%	1.62	5.22	5.23%	2.24
S8	3.01	3.24%	1.84	2.86	3.05%	2.12	7.50	7.86%	4.16
S9	2.90	2.90%	2.02	2.58	2.59%	1.75	7.46	7.22%	3.92
S10	2.59	2.56%	1.93	2.87	2.87%	1.96	6.67	6.34%	3.09
S11	3.55	3.83%	1.80	3.24	3.49%	1.97	6.43	6.67%	2.96
S12	1.79	1.75%	1.27	2.40	2.35%	1.46	6.32	5.99%	2.48
All	3.61	3.95%	2.13	4.00	4.44%	2.03	7.28	7.49%	3.82

**Table 3 sensors-21-02725-t003:** Mean absolute error ($AE_{mean}$) for SPARE, fastSPARE, and TROIKA by comparing heart rate error (BPM) from ECG (RR-intervals) and PPG (PP-intervals) on windows on 8 s and overlap of 2 s.

	Mean Absolute Error ${\mathrm{AE}}_{\mathrm{mean}}$ of HR Estimation (BPM)												
	Subj 1	Subj 2	Subj 3	Subj 4	Subj 5	Subj 6	Subj 7	Subj 8	Subj 9	Subj 10	Subj 11	Subj 12	Mean
SPARE	0.78	1.07	0.76	0.64	0.50	1.16	0.40	0.32	0.50	1.68	0.61	0.99	0.78
fastSPARE	1.05	1.13	0.65	0.82	0.46	0.49	0.36	0.33	0.27	1.67	0.54	0.91	0.72
TROIKA [23]	2.29	2.19	2.00	2.15	2.01	2.76	1.67	1.93	1.86	4.70	1.72	2.84	2.34
JOSS [51]	1.33	1.75	1.47	1.48	0.69	1.32	0.71	0.56	0.49	3.81	0.78	1.04	1.29

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
