# Peer review of "SPARE: A Spectral Peak Recovery Algorithm for PPG Signals Pulsewave Reconstruction in Multimodal Wearable Devices"

_sensors, 2021, doi:10.3390/s21082725_

Round 1

Reviewer 1 Report

The manuscript describes a novel method SPARE for the reconstruction of noisy blood pulse waves with motion artifact during daily life and physical activity, for heart-rate detection and other biomarkers. Two different approaches were proposed (SPARE and fastSPARE) with advantages according to the application. The paper is very interesting and the proposed method is well described with promising results. The technique is intended to be applied on wearable devices and requires an equipped ECG sensor. In order to improve the quality of the manuscript, the following concerns are suggested for authors:

Methods

Materials and methods are described throughout the text. The manuscript is mainly focused on the description of the technique. Also, the work does not include a data collection and three databases were used for its validation, including PPG and ECG signals. Although it was described in Section 5, an overall research design including those details (e.g. the availability of ECG data) would be desirable for an easy understanding of the manuscript. 

Results 

The results include equally spatiated data for sigma from 0 to 5000. How was that range defined? Also, Table 1 and Figures 9 and 10 presents the results equally spatiated for sigma. However, it was stated that sigma lower than 500 is closer to realistic MA for moderated physical activities. Maybe a semilog scale would be useful for a better understanding of lower values for sigma, which errors also are closer to the RAW signal.

Detection accuracy is shown in Table 1. Is it related to the average of windows (N according to equation 13 - 15) per subject? or to the results for all the participants in the databases?   

There is a lack of a statistical followed for assessing the results. In some parts, it is mentioned significant difference but it is not supported by a statistical test with a p-value. Also, some words like "like advantage"; "very low"; "better performance" should be justified by metrics. 

Minor concerns:

  • Line 135: the phrase "ECG data provided by the new wearable devices" can be confused since it seems to be referring to some specific device. Suggestion "...by new" instead "...by the new".
  • Line 162: Please, specify if the spectrum corresponds to the reconstructed signal. It would be desirable if it is also included in Figure 2.
  • I wonder about the spectrums from Figures 1, 2 and 4, which cover higher frequencies but most databases had sampling frequencies below 512 Hz. I suggest including if these signals were extracted from some database or were simulated.
  • Line 179: Please, include details about the band-pass filter and which platform was used for its implementation and as well as all the techniques.  
  • Figure 3, 5, 7: Please, include a definition for all the abbreviations used in the figures.
  • Line 319: The principal fiducial points and the biomarkers in Figure 7 should be cited also in the text.
  • Line 294: It seems that "p" is missing in the phrase "These are the set of peaks".
  • In Line 356 it was mentioned predictivity but in Line 425 it was specificity.
  • Line 426: Table 1 presents detection accuracy. It should be clarified.
  • Line 430: "baseline wander reduction technique" is a bit confused. It should be clear in the methodology.
  • Figure 9: Please use uniform concepts: "fastSPARE" instead "FFT-SPARE"; Legend: "RAW (orange line)" instead "without (orange line)"

Author Response

Please see the attachment reviewer 1.

Reviewer 2 Report

The paper entitled “SPARE: A SPectral peAk REcovery Algorithm for PPG Signals Pulsewave Reconstruction in Multimodal Wearable Devices” reports about an innovative algorithm able to detect and remove motion artefacts from PPG signals exploiting a simultaneous ECG signal. The paper is very interesting and well written. I suggest only few comments before publication:

  • Considering the importance of SPARE and fastSPARE for ecological applications in wearable devices, it is worth to discuss some possible solutions to decrease the computational time (it is stated that for 20 min long signals, 30 s are needed on a regular PC) and to implement the algorithms for a on-board real-time computation. Moreover, since the ECG signal is used to define the triplet to remove the motion artefacts, it is worth to consider that in wearable devices the ECG and PPG signals could be affected by similar motion artefacts, hence reducing the performances of the procedure. Please discuss these aspects in the discussion section.

  • In the figures there is not the unit of measure in the y-axis. Please specify the unit of measure or if they are normalized values.

  • In figure 7 the delay between the PPG and the ECG peaks is evident. The figure is quite misleading, because it seems that the ECG peaks are delayed with respect to the PPG peaks. In fact, the RR interval is highlighted between the first two ECG and PPG peaks, whereas the first RR interval should be coupled with the second PP interval. Please consider modifying the figure.

  • In the background some important references are missing:

Hina, A., & Saadeh, W. (2020). A noninvasive glucose monitoring SoC based on single wavelength photoplethysmography. IEEE transactions on biomedical circuits and systems, 14(3), 504-515.

Perpetuini, D., Chiarelli, A. M., Cardone, D., Rinella, S., Massimino, S., Bianco, F., ... & Merla, A. (2020). Photoplethysmographic prediction of the ankle-brachial pressure index through a machine learning approach. Applied Sciences, 10(6), 2137.

Sun, Z., He, Q., Li, Y., Wang, W., & Wang, R. K. (2021). Robust non-contact peripheral oxygenation saturation measurement using smartphone-enabled imaging photoplethysmography. Biomedical Optics Express, 12(3), 1746-1760.

Author Response

Please see the attachment reviewer 2.

Round 2

Reviewer 1 Report

I thank the authors for the review of the manuscript. I consider that all concerns were addressed and I recommend the article for publication on Sensors.